# Impact of M2BPGi on the Hepatocarcinogenesis after the Combination Therapy with Daclatasvir and Asunaprevir for Hepatitis C

**DOI:** 10.3390/biomedicines9060660

**Published:** 2021-06-08

**Authors:** Satoshi Takakusagi, Ken Sato, Kyoko Marubashi, Kazuko Kizawa, Takashi Kosone, Satoru Kakizaki, Hitoshi Takagi, Toshio Uraoka

**Affiliations:** 1Department of Gastroenterology and Hepatology, Kusunoki Hospital, Fujioka, Gunma 375-0024, Japan; satoshi.takakusagi@gmail.com (S.T.); mrbs1970kyoko@gmail.com (K.M.); kazukodr8715@yahoo.co.jp (K.K.); takashikosone@gmail.com (T.K.); htakagi@kusunoki-hp.com (H.T.); 2Department of Gastroenterology and Hepatology, Gunma University Graduate School of Medicine, Maebashi, Gunma 371-8511, Japan; kakizaki@gunma-u.ac.jp (S.K.); uraoka@gunma-u.ac.jp (T.U.)

**Keywords:** mac-2 binding protein glycosylation isomer, direct acting antivirals, liver fibrosis, hepatocarcinogenesis, sustained virological response

## Abstract

The clinical significance of mac-2 binding protein glycosylation isomer (M2BPGi) levels based on virological responses due to antiviral therapy has not been fully evaluated. We compared the change before and 24 weeks after the therapy with daclatasvir and asunaprevir (DCV+ASV) of M2BPGi levels with those of other fibrosis markers in 73 chronic hepatitis C cases. Moreover, we examined the association between M2BPGi levels and hepatocarcinogenesis in sustained virological response (SVR) and non-SVR cases. M2BPGi levels were significantly improved at post-treatment week 24 (PTW24) in SVR but not non-SVR cases, whereas the changes of other fibrosis markers showed the same tendency in both SVR and non-SVR cases. M2BPGi levels were well correlated with other fibrosis markers at baseline but not PTW24. The incidence of hepatocellular carcinoma (HCC) was significantly associated with M2BPGi levels at PTW24. The achievement of SVR significantly affected the improvement of M2BPGi levels that best reflected the effect of direct-acting antivirals among the fibrosis markers. Furthermore, M2BPGi levels at PTW24 were also associated with the incidence of HCC in only SVR cases. However, the rapid decrease of M2BPGi levels might reflect the amelioration of liver inflammation rather than the improvement of liver fibrosis, which should be further elucidated.

## 1. Introduction

In Japan, direct-acting antivirals (DAAs) have become the mainstream of chronic hepatitis C (CHC) treatment since the combination therapy with daclatasvir and asunaprevir (DCV+ASV) became available in 2014. In the most recent version (ver. 8.0) of the Japan Society of Hepatology (JSH) guideline for the management of DAA-naïve CHC without decompensated liver cirrhosis, combination therapies with sofosbuvir and ledipasvir (SOF/LDV), elbasvir and grazoprevir (EBR+GZR), glecaprevir and pibrentasvir (GLE/PIB) for genotype 1, and SOF and ribavirin (SOF+RBV), SOF/LDV, and GLE/PIB for genotype 2 have been recommended as the 1st line treatments, respectively. With these therapies, hepatitis C virus (HCV) eradications are possible in over 95% among treatment naïve patients [1,2,3,4,5,6].

On the other hand, the evaluation of liver fibrosis in patients with CHC is quite important as the absolute risk of hepatocellular carcinoma (HCC) occurrence that has been reported to be remained even after sustained virological response (SVR) in patients with advanced fibrosis, especially those with cirrhosis [7]. Although it has been well recognized that the histological analysis with liver biopsy is the gold standard for the evaluation of liver fibrosis, there are some limitations and problems such as sampling error and complication accompanying the procedure [8]. Laboratory methods such as aspartate aminotransferase (AST) to platelet ratio index (APRI) [9] and fibrosis-4 (FIB-4) index [10] have been suggested as non-invasive methods for liver fibrosis evaluation. However, these may be affected by factors other than liver diseases. Thus, the more specific marker is thought to be needed.

Glycomics-based markers for liver fibrosis were explored using a fully automated glycan-based immunoassay, FastLec-Hepa [11]. The mac-2 binding protein (M2BP) is a secretory glycoprotein that includes seven *N*-glycans per monomer [12]. A “doughnut-shaped” polymer that presents 70–112 *N*-glycans is formed by 10–16 monomers of M2BP in the serum [12]. M2BP is found to alter during the progression of liver disease and fibrosis due to the changes in N-glycosylation, which is sialylation or extension of polylactosamine, although the fundamental mechanism is unknown [12]. FastLec-Hepa identifies six lectin probes that could readily discriminate the altered *N*-glycans of M2BP and specifically binds to them and finally *Wisteria floribunda* agglutinin (WFA) is found to be the best candidate among six lectins at every fibrosis stage [11]. Thus, the altered *N*-glycans of M2BP is recognized by WFA specifically and this specific glycoprotein is described as WFA+-M2BP, which is renamed as M2BP glycosylation isomer (M2BPGi) [12]. A recent meta-analysis shows that M2BPGi could be a surrogate biomarker for liver biopsy to diagnose cirrhosis in chronic liver disease [12].

In addition, high M2BPGi levels were significantly associated with hepatocarcinogenesis and outcome in CHC cases with advanced fibrosis [13], and hepatocarcinogenesis in cases with the achievement of SVR [14,15,16]. Another glycomics-based marker, GlycoCirrhoTest identifies compensated cirrhosis patients at risk for HCC incidence [17]. Although the association between serum M2BPGi levels and the effect of DAAs has been reported [18,19,20,21], the change of serum M2BPGi levels in non-SVR cases has been insufficiently analyzed due to excellent DAAs efficacy.

In our university affiliated hospital, CHC patients with genotype 1b were treated by combination therapy with DCV+ASV as interferon (IFN) free first-commercially available DAAs combination therapy in Japan regardless of the presence of a Y93H substitution as pivotal non-structural (NS) 5A-resistance-associated substitutions (RASs) for the DAA-based therapy. The attending doctors in the hospital were obliged to use this therapy since the target patients included those with IFN-intolerance such as old age and advanced liver fibrosis and the information regarding the next-generation DAAs were not available in those days. Resultantly, a certain number of patients failed to achieve SVR in the hospital. Then, we compared the change of M2BPGi levels between the cases of SVR and non-SVR before and after the combination therapy with DCV+ASV. In addition, the correlations between M2BPGi levels and conventional established fibrosis markers, and change of α-fetoprotein (AFP) levels were analyzed. Furthermore, we also examined whether the association between M2BPGi levels and development of HCC after the combination therapy with DCV+ASV was affected by virological responses in those cases.

## 2. Materials and Methods

### 2.1. Patients

Seventy-three patients infected with HCV genotype 1b, who were treated with the combination therapy with DCV+ASV between 2014 and 2015 in our university affiliated hospital, were included in this analysis. The diagnosis of chronic HCV infection was based on the positivity of serum HCV RNA on the polymerase chain reaction (PCR) assay. Of the 73 patients, 37% (n = 27) were diagnosed as compensated liver cirrhosis based on laboratory data and imaging.

### 2.2. Treatment Regimen and Follow-up Surveillance for Viremia and HCC

Patients received 24 weeks of DCV (60 mg once daily) and ASV (100 mg twice daily) and were basically assessed as outpatients every 2 weeks during the treatment, and thereafter, followed-up every 4 weeks until 24 weeks after the combination therapy with DCV+ASV. After that, the consultation interval for each patient was determined at the discretion of the attending physician according to the condition of each patient. After the start of the combination therapy with DCV+ASV, patients had been examined with ultrasonography (US), contrast enhanced computed tomography (CT) or contrast enhanced magnetic resonance imaging (MRI) every 3 to 6 months until February 2020 for the surveillance of HCC. When questionable nodule (s) was (were) detected with US, additional investigation with contrast enhanced CT or MRI was performed for definite diagnosis of HCC.

### 2.3. Demographic and Laboratory Data

Demographic data such as sex, age, body mass index (BMI), previous history of HCC and IFN-based therapy, laboratory data such as platelet count, serum total bilirubin (T-bil), alanine aminotransferase (ALT), AST, albumin (Alb), and AFP levels were included as baseline data.

### 2.4. Virological Assessment

HCV RNA was measured at baseline and basically every 4 weeks after the start of the combination therapy with DCV+ASV using AccuGene m-HCV (Abbott Japan, Tokyo, Japan; lower limit of quantification, 1.1 log IU/mL). SVR at 24 weeks after the treatment (SVR24) was defined as continuously undetectable HCV RNA on the PCR assay until 24 weeks after the end of the combination therapy with DCV+ASV. Before the combination therapy with DCV+ASV, RASs in NS5A were investigated by the direct sequencing method (n = 26) or PCR-invader assay (n = 47). The direct sequencing method was performed by SRL (Tokyo, Japan) and the PCR-invader assay was performed by BML (Saitama, Japan).

### 2.5. Measurement of Mac-2 Binding Protein Glycosylation Isomer (M2BPGi) Levels

Serum M2BPGi levels were measured at baseline and post-treatment week 24 (PTW24) with the lectin-antibody sandwich immunoassay by the automated analyzer HISCL-5000 (Sysmex, Hyogo, Japan). The improvement of M2BPGi was defined as the statistically significant reduction of serum M2BPGi levels at post-treatment week 24 compared to the baseline.

### 2.6. Statistical Analysis

Continuous variables were expressed as median (interquartile range) and analyzed with the Mann-Whitney U test in unpaired analyses, and Wilcoxon signed-rank test in paired analyses. Categorical variables were analyzed with the Chi-squared test. The Spearman rank correlation coefficient test was used for the correlation of continuous variables. The cutoff of continuous variable was determined by the receiver operating characteristic (ROC) analysis. The cumulative incidence curve was determined and differences among the groups were assessed using Grey’s test. All statistical analyses were performed using EZR version 1.53 (Saitama Medical Center, Jichi Medical University, Saitama, Japan) [22].

## 3. Results

### 3.1. Baseline Characteristics of the Subjects

The baseline characteristics of 73 enrolled patients are shown in Table 1. The body mass index and the rates of Y93H RAS were significantly lower among SVR cases than non-SVR cases (*p* = 0.043, *p* = 0.04, respectively). There were no significant differences of age, sex, the rates of liver cirrhosis, the rates of patients with HCC history, pretreatment platelet count, serum T-bil, AST, ALT, Alb, AFP, HCV RNA, M2BPGi levels, APRI, FIB-4 index, and resistance-associated substitutions between SVR cases and non-SVR cases.

### 3.2. Change of Liver Fibrosis Markers at Baseline and PTW24

Serum M2BPGi levels significantly improved in SVR cases (3.27 cutoff index (COI) [1.49–5.13] at baseline vs. 1.48 COI [0.83–2.24] at PTW24, *p* < 0.001), while not in non-SVR cases (2.35 COI [1.65–5.75] at baseline vs. 2.47 COI [1.63–4.21] at PTW24, *p* = 0.107 (Figure 1A). On the other hand, APRI was significantly improved in both SVR cases (1.1 [0.8–2.2] at baseline vs. 0.7 [0.4–1.0] at PTW24, *p* < 0.01) and non-SVR cases (2.0 [0.9–2.9] at baseline vs. 1.0 [0.7–1.6] at PTW24, *p* < 0.05) (Figure 1B), and similarly, the FIB-4 index was also significantly improved in both SVR cases (3.67 [2.18–6.34] at baseline vs. 2.87 [2.47–4.76] at PTW24, *p* < 0.01) and non-SVR cases (4.87 [3.33–7.83] at baseline vs. 3.26 [2.14–4.67] at PTW24, *p* < 0.05) (Figure 1C). On the contrary, no significant differences of platelet counts were observed in both SVR cases (126 × 10^3^/μL [99 × 10^3^–167 × 10^3^] at baseline vs. 143 × 10^3^/μL [96 × 10^3^–185 × 10^3^] at PTW24, *p* = 0.243) and non-SVR cases (102 × 10^3^/μL [83 × 10^3^–152 × 10^3^] at baseline vs. 133 × 10^3^/μL [118 × 10^3^–187 × 10^3^] at PTW24, *p* = 0.229) (Figure 1D).

### 3.3. Correlations between M2BPGi Levels and Other Liver Fibrosis Markers That Are APRI, FIB-4 Index, and Platelet Counts

At baseline, serum M2BPGi levels showed significant correlations with APRI, FIB-4 index, and platelet counts (APRI: Spearman’s rank correlation coefficient (rs) = 0.79, FIB-4 index: rs = 0.78, and platelet count: rs = −0.68). On the other hand, no significant correlations between serum M2BPGi levels and other liver fibrosis markers were demonstrated at PTW24 (APRI: rs = 0.17, FIB-4 index: rs = 0.12, and platelet count: rs = 0.08, respectively) (Figure 2A–C). At PTW24, the correlations between serum M2BPGi levels and other liver fibrosis markers were also investigated separately in SVR and non-SVR cases. However, no significant correlations were observed in each analysis (APRI: rs = 0.18 in SVR cases, rs = −0.26 in non-SVR cases. FIB-4 index: rs = 0.17 in SVR cases, rs = −0.16 in non-SVR cases. Platelet count: rs = 0.04 in SVR cases, rs = 0.198 in non-SVR cases).

### 3.4. Change of AFP Levels at Baseline and PTW24

Serum AFP levels were significantly improved in both SVR cases (7.7 ng/mL [4.1–16.1] at baseline vs. 4.0 ng/mL [2.9–6.9] at PTW24, *p* < 0.001) and non-SVR cases (8.9 ng/mL [5.2–27.6] at baseline vs. 6.2 ng/mL [3.6–11.3] at PTW24, *p* < 0.05) (Figure 3).

### 3.5. The Association between M2BPGi Levels at PTW24 and Hepatocarcinogenesis

We evaluated the relationship between M2BPGi levels and HCC incidence after the combination therapy with DCV+ASV among the 55 patients without HCC history.

The median observation time (interquartile range) for HCC incidence from the start of the combination therapy with DCV+ASV was 1802 days (1085–1872). Eight (14.5%) of 55 patients had HCC incidence during the observation time. M2BPGi levels at PTW24 were significantly higher in the patients who developed HCC compared with the patients who had no HCC incidence (M2BPGi levels in the patients with HCC incidence: 2.42 COI [2.09–3.97] vs. M2BPGi levels in the patients without HCC incidence: 1.19 COI [0.81–2.36], *p* < 0.05) (Figure 4A). In ROC analysis, the best cutoff of M2BPGi levels at PTW24 regarding HCC incidence was 1.75 COI (sensitivity: 1.0, specificity: 0.66, positive predictive value: 0.333, negative predictive value: 1.0, diagnostic accuracy: 0.709) (Figure 4B). The area under the curve was 0.79 (95% confidence interval: 0.663–0.917) (Figure 4B).

In the cases with M2BPGi levels ≥1.75 COI at PTW24, the cumulative incidence rates of HCC at 1, 3, and 5 years were 0%, 22%, and 35.8%, respectively. On the other hand, in the cases with M2BPGi levels <1.75 COI at PTW24, there were no HCC incidence, and thus, the significant difference of the cumulative HCC incidence rates was demonstrated between the cases with M2BPGi levels ≥1.75 COI and those with M2BPGi levels <1.75 COI at PTW24 (*p* < 0.01) (Figure 5A).

Of the 55 cases without HCC history, 44 cases (80%) had achieved SVR. In SVR cases with M2BPGi levels ≥1.75 COI at PTW24, the cumulative incidence rates of HCC at 1, 3, and 5 years were 0%, 20%, and 40%, respectively, and the significant difference of the cumulative HCC incidence rates was demonstrated between the cases with M2BPGi levels ≥1.75 COI and those with M2BPGi levels <1.75 COI (*p* < 0.01) at PTW24 in SVR cases (Figure 5B). Regarding 11 of non-SVR cases, two (18.2%) cases had HCC incidence during the observation time. In non-SVR cases with M2BPGi levels ≥1.75 COI at PTW24, the cumulative incidence rates of HCC at 1, 3, and 5 years were 0%, 25%, and 25%, respectively. Although no cases with M2BPGi <1.75 at PTW24 had HCC incidence during the observation time, the statistical evaluation was difficult due to the small number of cases.

## 4. Discussion

Our most important finding is that M2BPGi levels significantly decreased in patients with SVR but not those without SVR. The other fibrosis markers, such as APRI, FIB-4 index, and platelet counts did not clearly reflect the efficacy of DCV+ASV. Thus, M2BPGi levels best reflected the effect of DAA-based therapy among these fibrosis markers. As the efficacy of DAA-based therapy is excellent, previous studies [18,19,20] could not fully evaluate the change of M2BPGi levels in non-SVR cases. In this respect, our study is of value.

The patients that achieved SVR after the DAA-based therapy showed the significant regression of liver stiffness assessed with transient elastography, APRI, and FIB-4 index [23], which was consistent with our findings that APRI and FIB-4 index improved significantly in SVR cases. However, those were improved significantly even in non-SVR cases (the statistical significance was numerically inferior compared with SVR cases). Although the exact reasons were unclear, a plausible reason is that the temporal improvement of platelet counts, serum AST levels, and serum ALT levels at PTW24 were achieved even in non-SVR cases. Actually, platelet counts increased and serum AST levels decreased in nine cases (60%) of non-SVR cases at PTW24. Serum ALT levels also decreased in 47% of non-SVR cases. These values might have affected the calculation formulas of FIB-4 index or APRI.

Importantly, M2BPGi levels do not always reflect only liver fibrosis. M2BPGi levels are increased at the time of the diagnosis in patients with acute liver injury and are significantly decreased at the time of ALT normalization [24,25]. The measurement intervals are average 280 days [24] and median 59 days [25]. M2BPGi levels reflect the severity and duration of liver injury, especially correlated with the time to ALT normalization in patients with acute liver injury [25]. Thus, M2BPGi levels can be drastically changed reflecting other causes such as liver inflammation in the relatively short term. Other glycomics-based markers may also be associated with liver inflammation. For example, a galactose-deficient anti-Gal immunoglobulin G [26] or GlycoFibroTest [27] that is the ratio between the agalacto glycans and the fully galactosylated glycans may reflect systemic inflammation including liver inflammation to some extent. As mentioned above, FAstLec-Hepa is a fully automated sandwich immunoassay used for direct quantitation of serum M2BPGi levels. The serum FAstLec-Hepa counts that mean serum M2BPGi levels decrease in SVR patients but not non-SVR patients by PEG-interferon-α/ribavirin therapy and FIB-4 index does not distinguish the differences of serum M2BPGi levels between SVR patients and non-SVR patients in a short post-therapeutic interval [11], which is a very similar finding to our results. In a long-term follow-up, out of three SVR patients, the serum FAstLec-Hepa counts numerically decrease at approximately 3 years after the completion of antiviral therapy compared to approximately 1 year after the completion of antiviral therapy in one case, while the counts fluctuate and do not show a continuous reduction from 1 to 5 years after the completion of antiviral therapy in the other cases [11]. Although the exact interpretation is difficult due to a few cases, the remarkable decrease of the counts in a short post-therapeutic interval might suggest amelioration of liver inflammation rather than improvement of liver fibrosis. We guess that this hypothesis is applicable to our results.

Before the combination therapy with DCV+ASV, serum M2BPGi levels significantly correlated with APRI, FIB-4 index, and platelet count. These results supported the usefulness of M2BPGi as a fibrosis marker. On the other hand, serum M2BPGi levels showed no significant correlations with APRI and FIB-4 index at PTW24. In SVR cases, the improvement degrees of APRI and FIB-4 index were numerically smaller than those of serum M2BPGi levels (Figure 2A–C). In non-SVR cases, the APRI and FIB-4 index were significantly decreased, while serum M2BPGi levels were not significantly changed. These changes might explain that serum M2BPGi levels were not correlated with the APRI and FIB-4 index at PTW24.

It was previously reported that SVR due to the combination therapy with DCV+ASV significantly contributed to the improvement of platelet counts [18]. In our study, platelet counts were improved in 53.4% (n = 31) in SVR cases, however, the significant improvement of platelet counts could not be obtained statistically. In our study, 58.9% (n = 43) were elderly patients who were 70 years old or more, and the median serum M2BPGi levels of overall patients before the combination therapy with DCV+ASV were relatively high values of 3.26 COI (1.58–5.28), thus, many established cirrhosis cases were thought to be included. Platelet counts might be improved significantly in the future study with long-term follow up and patients with milder liver fibrosis.

AFP is a non-invasive predictive marker for HCC in patients infected with HCV, which can be used as a complemental data for fibrosis stage [28]. In our study, serum AFP levels were significantly improved regardless of the achievement of SVR. Since serum AFP levels also reflect liver inflammation, the decrease of serum AFP levels in non-SVR cases might also be caused by the temporal improvement of hepatitis as with APRI and FIB-4 index. Similar to our study, it was reported that serum AFP levels were improved with the combination therapy with DCV+ASV even in non-SVR cases [29].

Although DAA-induced SVR was reportedly associated with the reduction of HCC risk, HCC incidence rates were higher in cirrhosis cases of SVR than non-cirrhosis cases of non-SVR [30]. In our study, M2BPGi levels at PTW24 were significantly high in patients who developed HCC after the treatment. The evaluation of M2BPGi levels at PTW24 could be useful in predicting the future HCC development. As the number of our study subjects was small and only one kind of DAA therapy was used in our study, the results were difficult to be generalized and should be interpreted carefully. However, the previous studies in which PTW12, PTW24 or 1 year after PTW24 is associated with HCC development in the patients treated with the combination therapy with DCV+ASV [19] or the regimens including other DAAs [14,31], support our data. M2BPGi levels after the time of ALT normalization might more accurately reflect the actual degree of liver fibrosis than that before the antiviral treatment by reducing the effect of liver inflammation or liver damage in CHC patients. Thus, it makes sense that M2BPGi levels after the time of ALT normalization might be a predictor of HCC development in CHC patients.

As mentioned above, the present study has several limitations. First, our study was limited to only Japanese patients infected with genotype 1 and treated with the combination therapy with DCV+ASV, although several DAA-based therapies replaced DCV+ASV at present in Japan. Second, this is the retrospective cohort study in a single institution. Third, the number of the study subjects was small and the follow up-period after the completion of the combination therapy with DCV+ASV was short.

## 5. Conclusions

In conclusion, M2BPGi levels significantly reduced in SVR cases but not in non-SVR cases. M2BPGi levels were well correlated with conventional established fibrosis markers before the combination therapy with DCV+ASV. Of these fibrosis markers that were APRI, FIB-4 index, and platelet counts, however, the change of these markers from baseline to PTW24 showed the same tendency between SVR and non-SVR cases. Thus, M2BPGi levels best reflected the effect of DAA-based therapy among these fibrosis markers. However, it should be remembered that the rapid decrease of M2BPGi levels might reflect amelioration of liver inflammation rather than improvement of liver fibrosis, which should be further elucidated. Importantly, M2BPGi levels at PTW24 were associated with the incidence ratio of HCC in only SVR cases. Further large-scale, longer follow-up, and prospective studies including other ethnic groups, genotypes, and DAAs regimens were needed to confirm our data.

## Figures and Tables

**Figure 1 biomedicines-09-00660-f001:**
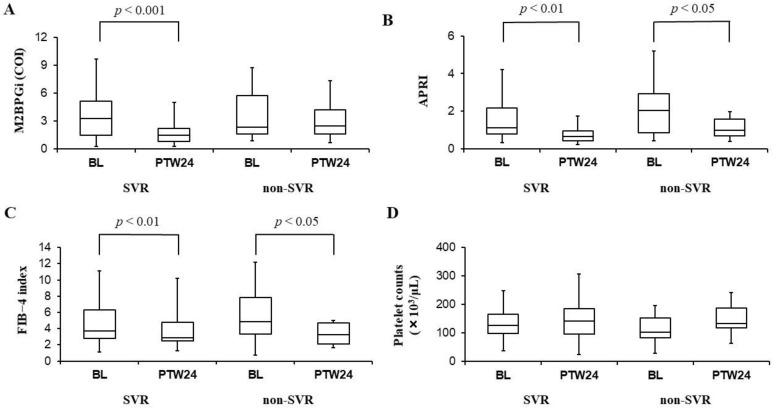
Serum mac-2 binding protein glycosylation isomer (M2BPGi) levels (**A**), aspartate aminotransferase to platelet ratio index (APRI) (**B**), fibrosis-4 (FIB-4) index (**C**), and platelet counts (**D**) before (BL: Baseline) and 24 weeks after the combination therapy with daclatasvir and asunaprevir (PTW24: Post-treatment week 24) in SVR and non-SVR cases. M2BPGi levels (**A**) significantly improved in SVR cases, while no significant differences of those levels were observed in non-SVR cases. APRI (**B**) and FIB-4 index (**C**) were significantly improved in both SVR and non-SVR cases. Platelet counts (**D**) had no significant changes in both SVR and non-SVR cases. In those box and whisker plots, the lines inside the boxes represent median values. Then, the upper and lower lines of boxes represent the 75th and 25th percentiles, respectively. Furthermore, the upper and lower bars outside the boxes represent the maximum and minimum values excluding outliers, respectively.

**Figure 2 biomedicines-09-00660-f002:**
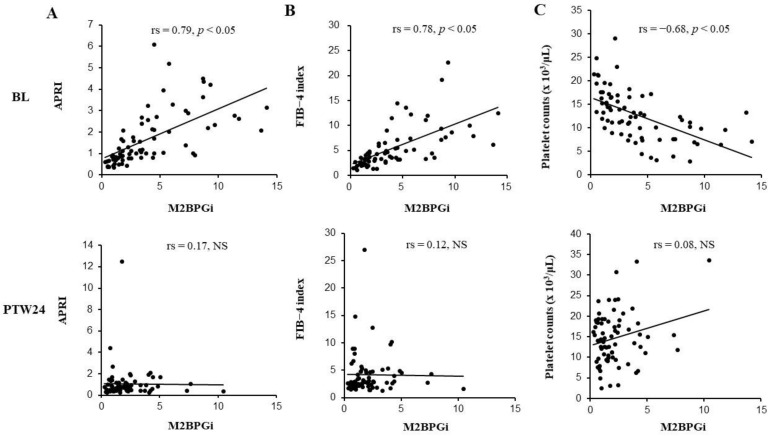
The correlation between mac-2 binding protein glycosylation isomer (M2BPGi) levels and APRI (**A**), FIB-4 index (**B**) or platelet counts (**C**). Before the combination therapy with daclatasvir (DCV) and asunaprevir (ASV), M2BPGi levels significantly correlated with APRI (**A**), FIB-4 index (**B**), and platelet counts (**C**), while these significant correlations disappeared 24 weeks after the combination therapy with DCV+ASV (PTW24: Post-treatment week 24). BL: Baseline; COI: Cutoff index; NS: Not significant; rs: Spearman’s rank correlation coefficient.

**Figure 3 biomedicines-09-00660-f003:**
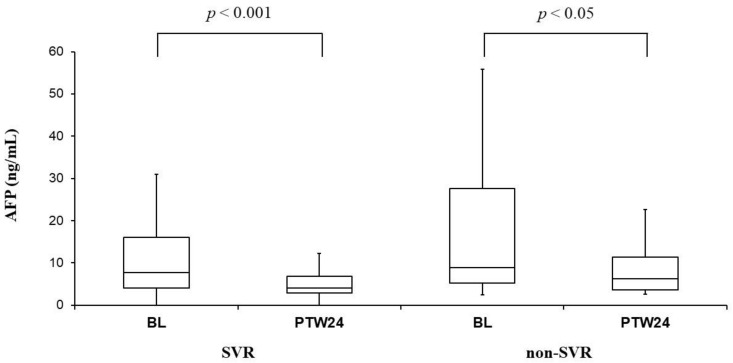
α-Fetoprotein (AFP) before (BL: Baseline) and 24 weeks after the combination therapy with daclatasvir (DCV) and asunaprevir (ASV) (PTW24: Post-treatment week 24) in sustained viral response (SVR) and non-SVR cases. Serum AFP levels significantly improved in both SVR and non-SVR cases. Those box and whisker plots were expressed as in Figure 1.

**Figure 4 biomedicines-09-00660-f004:**
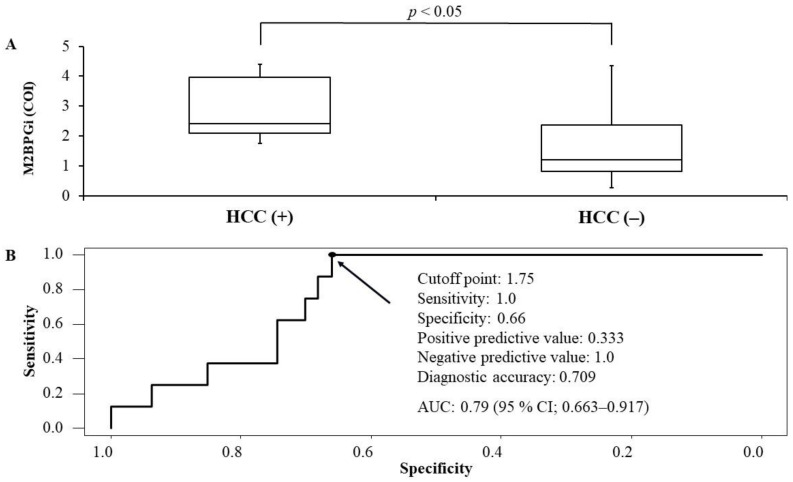
The comparison between mac-2 binding protein glycosylation isomer (M2BPGi) levels between patients who developed hepatocellular carcinoma (HCC) or not (**A**) and receiver operating characteristics analysis regarding the cutoff of M2BPGi levels for the development of HCC (**B**) after the start of daclatasvir + asunaprevir among the cases without HCC history. According to the Youden index, the best cutoff of M2BPGI levels was 1.75 (sensitivity: 1.0, specificity: 0.66, positive predictive value: 0.333, negative predictive value: 1.0, diagnostic accuracy: 0.709). Area under the curve was 0.79 (95% confidence interval: 0.663–0.917).

**Figure 5 biomedicines-09-00660-f005:**
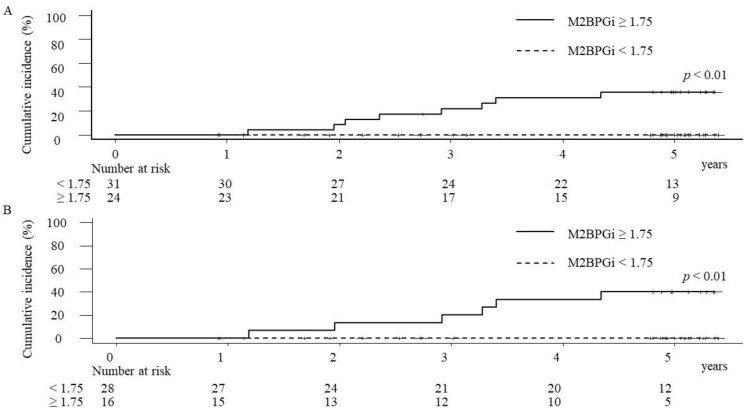
The cumulative hepatocellular carcinoma (HCC) incidence rates of overall cases (**A**) and SVR cases (**B**) after the start of the combination therapy with daclatasvir and asunaprevir. In both analyses of overall and SVR cases, the cumulative HCC incidence rates were significantly higher in the patients with the serum mac-2 binding protein glycosylation isomer (M2BPGi) ≥1.75 COI (cutoff index) than those with M2BPGi levels <1.75 COI.

**Table 1 biomedicines-09-00660-t001:** Baseline characteristics of study patients (n = 73).

	SVR (n = 58)	non-SVR (n = 15)	*p*
Sex (male/female)	25/33	10/5	0.18
Age (years)	73 (38–87) *	67 (50–86) *	0.17
Body mass index (kg/m^2^)	22.9 (21.6–25.2) *	25.3 (23.6–26.6) *	0.043
Chronic hepatitis/liver cirrhosis	39/19	7/8	0.141
History of HCC, n (%)	13 (22.4)	4 (26.7)	0.996
History of IFN treatments, n (%)	37 (63.8)	6 (40)	0.095
Platelet count (×10^3^/μL)	126 (99–167) *	102 (83–152) *	0.53
Total bilirubin (mg/dL)	0.6 (0.5–0.8) *	0.7 (0.6–1.1) *	0.072
AST (IU/L)	44 (34–57) *	50 (40–66) *	0.278
ALT (IU/L)	36 (27–50) *	46 (39–58) *	0.109
Albumin (g/dL)	4.2 (3.8–4.5) *	4.2 (3.8–4.8) *	0.547
AFP > upper normal limit (10 ng/mL), n (%)	20 (34.5)	7 (46.7)	0.141
HCV RNA (log IU/mL)	6.2 (5.8–6.6) *	6.4 (6.1–6.5) *	0.374
Mac-2 binding protein (COI)	3.27 (1.49–5.13) *	2.35 (1.65–5.75) *	0.881
APRI	1.1 (0.8–2.2) *	2.0 (0.9–2.9) *	0.309
FIB-4 index	3.67 (2.78–6.38) *	4.53 (2.81–7.7) *	0.482
Resistance-associated substitutions, n (%)			
L31V	1 (1.7)	2 (13.3)	0.197
Y93H	16 (27.6)	9 (60.0)	0.04

* Interquartile range. AFP: α-Fetoprotein; ALT: Alanine aminotransferase; APRI: Asparate aminotransferase to platelet ratio index; AST: Aspartate aminotransferase; COI: Cutoff index; Fib-4: Fibrosis 4 index; HCC: Hepatocellular carcinoma; HCV: Hepatitis C virus; IFN: Interferon.

## Data Availability

The analyzed datasets in the study are available from the corresponding author on reasonable request.

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
