# Peer review of "Impact of M2BPGi on the Hepatocarcinogenesis after the Combination Therapy with Daclatasvir and Asunaprevir for Hepatitis C"

_biomedicines, 2021, doi:10.3390/biomedicines9060660_

Round 1

Reviewer 1 Report

In this manuscript two scientific questions are addresses:

  • is M2BPGi related to presence of SVR in HCV patients?
  • is M2BPGi related to the risk of HCC development

The cohort is limited to Japanese patients with genotype 1 treated with DCV+ASV

It is important to address both questions apart of each other. I have some concerns on design, and severe concerns regarding the conclusions. 

1/ Design: the cohort is very specific and the number of patients is quite low, what makes it difficult to generalise the results. 

2/ The approach of using non-invasive biomarkers to monitor treatment response is interesting and promising. This is another illustration that non-invasive bimoarkers change rapidly upon decrease of virale load in HCV treatment. However, the conclusion that rapide decrease of fibrosis in SVR appears during or early after initiation of DAA in HCV is not correct. We now that the same is observed in elastometry levels, and it is well known that this does not reflect regression of fibrosis, what could not happen at this pace, but a regression of inflammation, leading to edema, leading to increase elastometry levels. This brings me to the next point, this is probably not a marker of fibrosis, but a marker of inflammation. 

We know that other glycomic markers of fibrosis are not markers of fibrosis, but rather markers of inflammation, what explains the dynamic changes observed in this kind of situations, I refer to 3 papers that should be cited in this discussion

  • Mehta AS, Long RE, Comunale MA, et al. Increased
    levels of galactose-deficient anti-Gal immunoglobulin G
    in the sera of hepatitis C virus-infected individuals with
    fibrosis and cirrhosis. J Virol 2008;82:1259–1270.
  • Kuno A, Ikehara Y, Tanaka Y, et al. A serum “sweetdoughnut” protein facilitates fibrosis evaluation and therapy assessment in patients with viral hepatitis. Sci Rep 2013;3:1065.
  • Vanderschaeghe D, Laroy W, Sablon E, et al. GlycoFibroTest
    is a highly performant liver fibrosis biomarker derived from DNA sequencer-based serum protein glycomics. Mol Cell Proteomics 2009;8:986–994

To provide more understanding in this marker, the authors should give more information on the characteristics of this biomarker and the underlying glycosylation related changes that are measured. 

Also, the use of glycomics-based markers to assess the risk of HCC development is not new. 

Relevant papers to cite are

- Wisteria floribunda Agglutinin-Positive Mac-2 Binding Protein but not alpha-fetoprotein as a Long-Term Hepatocellular Carcinoma Predictor.

Osawa L, Tamaki N, Kurosaki M, Kirino S, Watakabe K, Wang W, Okada M, Shimizu T, Higuchi M, Takaura K, Takada H, Kaneko S, Yasui Y, Tsuchiya K, Nakanishi H, Itakura J, Takahashi Y, Enomoto N, Izumi N. Int J Mol Sci. 2020 May 21;21(10):3640. doi: 10.3390/ijms21103640.

and

Verhelst X, Vanderschaeghe D, Castera L, et al. A glycomics-based test predicts the development of hepatocellular carcinoma in cirrhosis. Clin Cancer Res 2017;23:2750–2758.

Including this information would increase the quality of the interpretation of these results. 

Author Response

Reviewer 1's comments:

In this manuscript two scientific questions are addresses:

is M2BPGi related to presence of SVR in HCV patients?

is M2BPGi related to the risk of HCC development

The cohort is limited to Japanese patients with genotype 1 treated with DCV+ASV

It is important to address both questions apart of each other. I have some concerns on design, and severe concerns regarding the conclusions.

1/ Design: the cohort is very specific and the number of patients is quite low, what makes it difficult to generalise the results.

We totally agree with you. As you mentioned, the cohort is very specific and the number of patients is quite low. DCV/ASV therapy was first introduced into Japan as Interferon-free DAA therapy. The efficacy of DCV/ASV therapy is inferior to that of the following DAA therapies because the efficacy is extremely low for patients with resistance-associated substitutions. In addition, in our university affiliated hospital not but another hospital, DCV/ASV therapy was dared performed for these intractable patients. Resultantly, not a few non-SVR patients were observed. In general, very few non-SVR patients could be observed in treatment-naïve patients with the current DAA therapies because the efficacy was excellent (more than 95%). The important aim of our study was to compare the clinical significance of M2BPGi levels between SVR patients and non-SVR patients. That is why we focused on the single institution and DCV/ASV therapy. As you suggested, we cannot deny week points. Then, we added references #14 that you recommended to cite and #31 and discussed the association between M2BPGi protein and hepatocarcinogenesis in chronic hepatitis C patients who received the regimens including other DAA to strengthen our result as the reviewer 2 suggested, because this is the most important point of our manuscript. Although PEG-interferon-α/ribavirin therapy but not DAA therapy was performed, our results regarding the dynamics of M2BPGi protein levels were very similar to the results of the study (Ref #11) that you also recommended to cite. Although we also cited this article as Ref #11 in the original manuscript, we did not discuss in this regards. Thanks to you, subsequently we believe that the reliability of our data could be increased. Besides, we underscored this weak point again at lines 311-313 on page 11. We would like to extend a special thank you because we could increase the quality of reliability of our results due to your shrewd advice.

2/ The approach of using non-invasive biomarkers to monitor treatment response is interesting and promising. This is another illustration that non-invasive bimoarkers change rapidly upon decrease of virale load in HCV treatment. However, the conclusion that rapide decrease of fibrosis in SVR appears during or early after initiation of DAA in HCV is not correct. We now that the same is observed in elastometry levels, and it is well known that this does not reflect regression of fibrosis, what could not happen at this pace, but a regression of inflammation, leading to edema, leading to increase elastometry levels. This brings me to the next point, this is probably not a marker of fibrosis, but a marker of inflammation.

We know that other glycomic markers of fibrosis are not markers of fibrosis, but rather markers of inflammation, what explains the dynamic changes observed in this kind of situations, I refer to 3 papers that should be cited in this discussion

Mehta AS, Long RE, Comunale MA, et al. Increased

levels of galactose-deficient anti-Gal immunoglobulin G

in the sera of hepatitis C virus-infected individuals with

fibrosis and cirrhosis. J Virol 2008;82:1259–1270.

Kuno A, Ikehara Y, Tanaka Y, et al. A serum “sweetdoughnut” protein facilitates fibrosis evaluation and therapy assessment in patients with viral hepatitis. Sci Rep 2013;3:1065.

Vanderschaeghe D, Laroy W, Sablon E, et al. GlycoFibroTest

is a highly performant liver fibrosis biomarker derived from DNA sequencer-based serum protein glycomics. Mol Cell Proteomics 2009;8:986–994

We cited 3 papers that you mentioned as ref #11(originally ref #11), 26, 27 and one more paper as ref #25 in addition to #24(originally ref #21) that was missing in the original manuscript by mistake. Then, we described that these other glycomic markers of fibrosis may be associated with inflammation in the “Discussion” section. Please see lines 258-281 on Page 10,11. Taken together, we corrected the conclusions, in which the rapid decrease of M2BPGi protein levels might reflect amelioration of liver inflammation rather than improvement of liver fibrosis. Please see lines 333-336 on Page 12. We also changed the “Abstract” to reflect this content.

To provide more understanding in this marker, the authors should give more information on the characteristics of this biomarker and the underlying glycosylation related changes that are measured.

As you suggested, we added more information on the characteristics of M2BPGi and the underlying glycosylation related changes that are measured by citing ref #12 and #11 that you recommended to cite. Please see lines 54-67 on page 2.

Also, the use of glycomics-based markers to assess the risk of HCC development is not new.

Relevant papers to cite are

- Wisteria floribunda Agglutinin-Positive Mac-2 Binding Protein but not alpha-fetoprotein as a Long-Term Hepatocellular Carcinoma Predictor.

Osawa L, Tamaki N, Kurosaki M, Kirino S, Watakabe K, Wang W, Okada M, Shimizu T, Higuchi M, Takaura K, Takada H, Kaneko S, Yasui Y, Tsuchiya K, Nakanishi H, Itakura J, Takahashi Y, Enomoto N, Izumi N. Int J Mol Sci. 2020 May 21;21(10):3640. doi: 10.3390/ijms21103640.

and

Verhelst X, Vanderschaeghe D, Castera L, et al. A glycomics-based test predicts the development of hepatocellular carcinoma in cirrhosis. Clin Cancer Res 2017;23:2750–2758.

Including this information would increase the quality of the interpretation of these results.

We added these two papers that you suggested as ref #14 and #17 in the “Introduction” section regarding the use of glycomics-based markers to assess the risk of HCC development. Please see lines 68-71 on page 2.

Reviewer 2 Report

This is a well-written article about Mac-2 binding protein glycosylation isomer (M2BPGi) protein, a serological glycobiomarker evaluating liver fibrosis after direct-acting antiviral DAA therapy. Even though the specific DAA therapy (daclatasvir and asunaprevir) is no longer in use (mentioned by the authors as one of the limitations of the study), the results of the study are interesting especially in the context of hepatocarcinogenesis. The study is well designed, with two groups (SVR and non-SVR) and additional long-term follow-up regarding the cumulative incidence of HCC. Data in the manuscript is clearly presented, and tables and graphs are self-explanatory.

I would recommend publication of the article after resolving several issues:

Please explain lines 83/84 …” All patients were diagnosed as Child-Pugh class A; however, 37% (n = 27) were diagnosed as compensated liver cirrhosis based on laboratory data and imaging”… Child-Pugh class A is compensated cirrhosis, so what do authors consider by 37%?

I would strongly recommend adding the discussion on M2BPGi protein in the context of other current DAA therapies and elaborating the part regarding M2BPGi protein and hepatocarcinogenesis.

Author Response

Reviewer 2's comments:

This is a well-written article about Mac-2 binding protein glycosylation isomer (M2BPGi) protein, a serological glycobiomarker evaluating liver fibrosis after direct-acting antiviral DAA therapy. Even though the specific DAA therapy (daclatasvir and asunaprevir) is no longer in use (mentioned by the authors as one of the limitations of the study), the results of the study are interesting especially in the context of hepatocarcinogenesis. The study is well designed, with two groups (SVR and non-SVR) and additional long-term follow-up regarding the cumulative incidence of HCC. Data in the manuscript is clearly presented, and tables and graphs are self-explanatory.

I would recommend publication of the article after resolving several issues:

Thank you very much for your comments.

Please explain lines 83/84 …” All patients were diagnosed as Child-Pugh class A; however, 37% (n = 27) were diagnosed as compensated liver cirrhosis based on laboratory data and imaging”… Child-Pugh class A is compensated cirrhosis, so what do authors consider by 37%?

We apologize. As you suggested, the description was inappropriate. The liver function of chronic hepatitis as well as liver cirrhosis was evaluated and described as Child-Pugh class A. Originally, the classification is applicable to liver cirrhosis only. We corrected the corresponding sentences. Please see the lines 94-96 on page 2.

I would strongly recommend adding the discussion on M2BPGi protein in the context of other current DAA therapies and elaborating the part regarding M2BPGi protein and hepatocarcinogenesis.

We added references #14, #31 and discussed the association between M2BPGi protein and hepatocarcinogenesis in chronic hepatitis C patients who received the regimens including other DAAs. Please see the lines 313-316 on page 11.

Round 2

Reviewer 1 Report

All remarks have been addressed well.